# A Facile Method to Fabricate Al_2_O_3_-SiO_2_ Aerogels with Low Shrinkage up to 1200 °C

**DOI:** 10.3390/molecules28062743

**Published:** 2023-03-17

**Authors:** Yulin Tian, Xiaodong Wang, Yu Wu, Xiaoxue Zhang, Chun Li, Yijun Wang, Jun Shen

**Affiliations:** Shanghai Key Laboratory of Special Artificial Microstructure Materials and Technology, School of Physics Science and Engineering, Tongji University, Shanghai 200092, China

**Keywords:** Al_2_O_3_-SiO_2_ composite aerogels, high-temperature performance, high specific surface area

## Abstract

Monolithic Al_2_O_3_-SiO_2_ composite aerogels were synthesized by using inexpensive aluminum chloride hexahydrate (AlCl_3_·6H_2_O) and tetraethyl orthosilicate (TEOS). By adjusting the molar ratio of Al and Si, the best ratio of high-temperature resistance was found. The resultant aerogels (Al:Si = 9:1) exhibit high thermal performance, which can be identified by the low linear shrinkage of 5% and high specific surface area (SSA) of 283 m^2^/g at 1200 °C. Alumina in these aerogels mainly exists in the boehmite phase and gradually transforms into the θ-Al_2_O_3_ phase in the process of heating to 1200 °C. No α-Al_2_O_3_ is detected in the heating process. These Al_2_O_3_-SiO_2_ composite aerogels are derived from a simple, low-priced and safe method. With their high thermal performance, these aerogels will have a wide application in high-temperature field.

## 1. Introduction

Alumina aerogels are unique nanoporous materials with low thermal conductivity, high specific surface area and high temperature performance [1,2,3,4,5,6]. Due to the excellent performance at high temperature, alumina aerogels were considered to be high-temperature insulation materials and high-temperature catalysts. In addition, alumina aerogels were used as catalysts for the reactions of NO reduction, CO oxidation, and hydrocarbons oxidation. It should be noted that alumina aerogels retain a high specific surface area up to 800 °C, which is very important for catalytic reaction since most of them occur at elevated temperatures [7,8,9]. However, the sintering and brittleness of alumina aerogels at high temperatures have always been important factors hindering their practical application [10,11,12,13]. With the development of thermal insulation materials, it was found that the alumina aerogels obtained by doping and composite have greatly improved high-temperature thermal stability, which gives the alumina-based aerogels better application prospects [14,15,16].

Alumina aerogels usually use aluminum alkoxides or aluminum inorganic salts as precursors, adding Si, Zr, Ti, Y and La to improve their high-temperature thermal stability. The precursors of alumina aerogels are mainly divided into alumina alkoxides such as sec-butoxide (ASB) or isopropoxide (AIP), and alumina inorganic salts such as aluminum chloride (AlCl_3_·6H_2_O) or aluminum nitrate (Al(NO_3_)_3_·9H_2_O) [17]. For the alumina alkoxides, Yang et al. prepared silica-doped alumina aerogels via ASB and TEOS as precursors by rapid supercritical drying, which exhibited a stable γ-Al_2_O_3_ phase and a high SSA of 146 m^2^/g at 1200 °C when the silica content was in the range of 10.6–13.1 wt% [18]. Wang et al. modified the alumina aerogels derived from ASB using trimethylethoxysilane (TMEO), the modified samples exhibited no shrinkage, a θ-Al_2_O_3_ phase and a high SSA of 147 m^2^/g at 1200 °C [19]. Zhang et al. synthesized alumina aerogels from AlCl_3_·6H_2_O using acetoacetic-grafted polyvinyl alcohol as a template; the samples possessed high elastic modulus of 8.81 MPa, a γ-Al_2_O_3_ phase and a high SSA of 143 m^2^/g at 1000 °C [20]. Although the alumina aerogels derived from alumina alkoxides exhibited good thermal performance, the alumina alkoxides demonstrate ultrahigh reactivity, complex chemical reaction pathways, unacceptable price and biological safety issues, which become huge obstacles to high-temperature applications [21]. The method of using inorganic aluminum salts as the precursor to prepare alumina aerogels is simple, and the precursor cost is low, so it has a better industrial application value. The existing research results show that introducing SiO_2_ into alumina aerogels is an effective and economical way to improve the high-temperature performance [22,23].

In this work, we applied AlCl_3_·6H_2_O and TEOS to prepare Al_2_O_3_-SiO_2_ composite aerogels using the propylene oxide (PO) addition method. The method uses PO as a proton-trapping agent to promote the multistage hydrolysis of Al^3+^ and form a stable three-dimensional network structure [24]. In general, adding SiO_2_ into alumina aerogels will enhance the thermal stability of the gel at high temperature. However, the introduction of too much SiO_2_ may reduce the high-temperature thermal performance of alumina aerogels. In order to find out the range of SiO_2_ content, which can enhance the performance of alumina aerogels, we prepared several groups of samples from low to high SiO_2_ content. Among them, samples with lower SiO_2_ content showed better thermal stability than pure alumina aerogels, while samples with higher SiO_2_ content experienced reduced high-temperature performance. For samples with low SiO_2_ content, the introduction of SiO_2_ enhances the skeleton strength and mechanical strength of alumina aerogels and also suppresses the phase transition from θ-Al_2_O_3_ to α-Al_2_O_3_, thus improving the thermal properties [25]. This work studied the influence of silicon content on the high-temperature performance of aerogels in Al_2_O_3_-SiO_2_ composite aerogels and found the range of SiO_2_ that provides a positive effect. These findings have a guiding role in the subsequent Al_2_O_3_-SiO_2_ composite process. These Al_2_O_3_-SiO_2_ composite aerogels with high thermal performance derived from a simple, low-priced and safe method have a wide application in the high-temperature field.

## 2. Results

### 2.1. Synthesis of Al_2_O_3_-SiO_2_ Aerogels

Al_2_O_3_-SiO_2_ composite aerogels were prepared by the PO addition method and ethanol supercritical drying process [24]. AlCl_3_∙6H_2_O and TEOS were dissolved in 84 mL of ethanol aqueous solution (the volume ratio of ethanol to water was 7:3), and 35mL of PO was added after stirring for ten minutes. The total amount of AlCl_3_∙6H_2_O and TEOS was 0.05 mol. The samples were denoted as AS0, AS1, AS2, AS3, AS4 and AS5 according to the molar ratios of TEOS to AlCl_3_∙6H_2_O = 0:10, 1:9, 2:8, 3:7, 4:6 and 5:5, respectively. After aging for one day, the solvent of wet gel was displaced with fresh ethanol 7 times (12 h for each time). The aerogels were finally obtained by the ethanol supercritical drying process with temperature of 265 °C and pressure of 10 MPa. Then the samples were heat-treated at 600, 800, 1000 and 1200 °C for 1 h gradually, and the corresponding samples were denoted by adding the heat treatment as a suffix (e.g., AS0-600 and AS1-1200).

### 2.2. Macro Morphology and Properties at Room Temperature

Figure 1 shows the macroscopic morphology of Al_2_O_3_-SiO_2_ composite aerogels. With the increase in SiO_2_ content, the color of these aerogels gradually deepened, and the sample cracked when the molar ratio of Si to Al was 1:1. The density, linear shrinkage and thermal conductivity of the aerogels are shown in Table 1. The linear shrinkage here refers to the reduction in diameter from wet gel to aerogel. The linear shrinkage of AS1 and AS2 is lower than that of AS0, which indicates that the adding of a small amount of SiO_2_ improves the thermal stability of the aerogels. The linear shrinkage of AS3, AS4 and AS5 is significantly higher than that of AS0, which indicates the excessive adding of SiO_2_ will not only destroy the formability of alumina aerogel bulks but also reduce their thermal stability. The thermal conductivities of AS samples at room temperature (20 °C) are between 0.033 and 0.037 W/m∙K, indicating that AS samples have excellent thermal insulation ability.

### 2.3. Shrinkage after Different Heating Temperatures

Figure 2a shows the linear shrinkage of AS samples under high-temperature heat treatment, which can represent their high-temperature thermal stability. Figure 2b shows the physical photos of AS1 sample before and after treatment at 1200 °C. Compared with AS0, the linear shrinkage of AS1 and AS2 between 1000 °C and 1200 °C is significantly reduced, which indicates that adding a small amount of silicon can effectively improve the high-temperature thermal stability of alumina-based aerogels. The high-temperature linear shrinkage of AS4 and AS5 is significantly larger than that of AS0, which is due to the introduction of a large amount of SiO_2_, and the SiO_2_ starts to agglomerate at a high temperature above 600 °C. Among them, the linear shrinkage of AS1 after heat treatment at 1200 °C is only 5%, which shows excellent thermal stability at high temperature.

According to the different amount of silica, the AS samples transformed to γ-, δ-, θ-Al_2_O_3_ and mullite [15,25]. In the process of sol gel, silica is basically amorphous, and alumina exists as the boehmite phase [21]. Finally, SiO_2_ is introduced to form -Si-O-Al- bonds, and the Si distributed in the alumina lattice will inhibit the phase transition [26]. Therefore, increasing the uniform distribution of SiO_2_ in alumina aerogels is conducive to improving the thermal stability of alumina aerogels.

### 2.4. Micromorphology at Different Temperatures

Figure 3 shows the microstructure of AS0 from 20 to 1200 °C. As shown in Figure 3a–e, the alumina aerogel AS0 exhibits a fibrous three-dimensional network porous structure, which is mainly composed of lamellar and strip nanoparticles. With the increment of heat treatment temperature, there is no obvious particle aggregation and structure collapse, and AS0 still maintains the nanoporous structure. As the temperature rises to 1200 °C, the nanoporous structure of AS0 does not change, but it can be seen that the voids are significantly smaller and the structure is more compact. Figure 4 shows the microstructure of AS1 from 20 to 1200 °C. Compared with the AS0 sample, the structure of AS1 is obviously different and mainly composed of larger lamellar and strip nanoparticles. After 1200 °C heat treatment, the structure of AS1 does not change significantly. It can be considered that the introduction of SiO_2_ makes the microstructure of alumina aerogels coarser. At the same time, the introduction of SiO_2_ can reduce the hydroxyl groups on the surface of Al_2_O_3_, prevent the dehydroxylation reaction and thus inhibit its phase transformation at high temperature [27,28]. In addition, SiO_2_ can also reduce the contact between Al_2_O_3_ particles and effectively control the surface/bulk diffusion, thus improving the temperature resistance of Al_2_O_3_ aerogel [29].

### 2.5. Variation of Chemical Composition at Different Temperatures

Figure 5a, b show the variation in chemical bonds of the AS0 and AS1 samples after heat treatment from 20 to 1200 °C. As shown in Figure 5a, AS0 mainly presents as the boehmite phase at room temperature after the supercritical drying process. The peaks at 3449 and 1634 cm^−1^ correspond to the stretching and bending vibrations of the physically adsorbed water, respectively. The bands at 1075 cm^−1^ correspond to the vibration modes of Al-OH in the boehmite crystal phase. The bands at 885, 780, 620 and 489 cm^−1^ also correspond to the vibration mode of Al-O-Al and torsional vibration, stretching vibration and bending vibration of Al-O in the boehmite crystal [30,31,32], respectively. When the heat-treated temperature rises above 600 °C, the characteristic infrared peaks corresponding to the boehmite phase of AS0 all disappear, indicating that the crystal phase transition of alumina aerogels has taken place. All these absorption bands of the pseudo-boehmite structure disappear after heat treatment at 600 °C, followed by the emergence of broad adsorption bands in the low-frequency region of 500–900 cm^−1^ [30,31,32]. With the further increment of temperature, the absorption band of Al_2_O_3_ aerogel becomes wider, indicating that its crystallinity is improved. After heat treatment at 800 °C, there are two wide absorption peaks at 574 and 829 cm^−1^, which correspond to the vibration modes of Al-O in the θ-Al_2_O_3_ phase, indicating the appearance of a partial θ-Al_2_O_3_ crystal phase. In the process of heat treatment, the characteristic absorption peaks of 448 and 637 cm^−1^ corresponding to α-Al_2_O_3_ do not appear all the time, indicating that there is no α-Al_2_O_3_ crystal phase, and the alumina AS0 sample shows good thermal stability [30]. As shown in Figure 5b, the Al_2_O_3_-SiO_2_ aerogel AS1 also presents as the boehmite crystal phase at room temperature. When the heat-treated temperature rises to 600 °C, the characteristic infrared peaks corresponding to the boehmite phase of AS1 also disappear, and a wide absorption band corresponding to the γ-Al_2_O_3_ crystal appears. The infrared peaks of AS1 hardly change in the subsequent heating process, indicating that the addition of SiO_2_ effectively inhibits the phase transition of alumina aerogels and keeps the alumina aerogels intact, which is consistent with the previous linear shrinkage of only 5% at 1200 °C.

Figure 5c reflects the changes of crystal structure of alumina aerogel AS0 after heat treatment from 20 to 1200 °C. The AS0 sample shows its boehmite phase at room temperature. The sharp peaks at 2θ = 28°, 38°, 49°, 65° and 72° correspond to the characteristic diffraction of boehmite (120), (031), (200), (002) and (251) crystal planes [33]. After heat treatment at 600 °C, AS0 transforms from the boehmite phase to the γ-Al_2_O_3_ phase. The peaks at 2θ = 46° and 66° correspond to the diffraction of γ-Al_2_O_3_ (400) and (440) crystal planes, respectively [34]. However, the peaks of other γ-Al_2_O_3_ crystal planes do not appear, indicating that the crystallinity of the γ-Al_2_O_3_ phase is low. After heat treatment at 800 °C, the peaks of the γ-Al_2_O_3_ crystal phase are more obvious, and new weak peaks at 2θ = 37° and 39° appear, corresponding to the diffraction peaks of γ-Al_2_O_3_ (311) and (222) crystal planes, respectively [35]. After heat treatment at 1000 °C, γ-Al_2_O_3_ transforms into the θ-Al_2_O_3_ phase, and the peaks at 2θ = 39°, 45° and 67° correspond to the diffraction of θ-Al_2_O_3_ (104), (21-1) and (215), respectively. After heat treatment at 1200 °C, the characteristic diffraction peaks of the θ-Al_2_O_3_ phase become stronger, and a new peak at 2θ = 33° that corresponds to the (20-2) crystal plane diffraction of the θ-Al_2_O_3_ phase appears, indicating that AS0 exhibits a stable θ-Al_2_O_3_ crystal phase and does not transform into an ɑ-Al_2_O_3_ phase at 1200 °C. Figure 5d reflects the changes of crystal phase of Al_2_O_3_-SiO_2_ composite aerogels AS1 after heat treatment at different temperatures. The AS1 sample also shows the same boehmite crystal structure and has five characteristic diffraction peaks at 2θ = 28°, 38°, 49°, 65° and 72° at room temperature. After heat treatment at 600 and 800 °C, AS1 also transforms into the γ-Al_2_O_3_ phase, but its characteristic diffraction peaks only appear at 2θ = 66°, indicating the overall crystallinity is very low. After heat treatment at 1000 and 1200 °C, AS1 transforms into the θ-Al_2_O_3_ crystal phase, and the characteristic diffraction peak is also reduced; only peaks at 2θ = 45° and 67° appear, which indicates that the addition of SiO_2_ can effectively hinder the crystal transition of alumina aerogels and improve their thermal stability significantly.

Figure 6a reflects the infrared absorption peaks of all AS samples after calcination at 1200 °C for 1 h. The absorption peaks at 3449 and 1634 cm^−1^ correspond to the stretching and bending vibration of physically adsorbed water, respectively. The bands at 570 and 830 cm^−1^ correspond to the vibration mode of Al-O in the θ-Al_2_O_3_ phase. With the increment of the Si/Al molar ratio, AS samples exhibit different FTIR spectra. After heat treatment at 1200 °C, AS samples at 570 and 830 cm^−1^ also contain weak absorption peaks, indicating that they are mainly in the θ-Al_2_O_3_ crystal phase, but the crystallization degree is lower than that of AL samples. In addition, the bands at 467 and 1094 cm^−1^ in AS4 and AS5 correspond to the vibration modes of Si-O-Si, while these absorption peaks do not appear in the samples with low silica content such as AS1, AS2 and AS3, indicating that silicon mainly exists in the form of Al-O-Si in these samples, which is helpful to improving the thermal stability of alumina aerogels.

Figure 6b demonstrates the crystal phase of all the AS samples after heat treatment at 1200 °C. For Al_2_O_3_-SiO_2_ composite aerogels, AS1, AS2 and AS3 only show a θ-Al_2_O_3_ crystal phase, but the crystallinity is lower than that of AS0, which indicates that the introduction of SiO_2_ effectively hinders the crystal phase transition of alumina aerogels. However, too much SiO_2_ exhibits adverse effects. In addition to the characteristic diffraction peaks of the θ-Al_2_O_3_ crystal phase, the AS4 sample also has the characteristic diffraction peaks of ɑ-Al_2_O_3_ at 2θ = 26°, 35°, 53° and 58°, which correspond to ɑ-Al_2_O_3_ (012), (104), (113) and (024) crystal planes, respectively [11,27]. Therefore, the AS4 sample presents a mixed crystal phase of θ-Al_2_O_3_ and α-Al_2_O_3_, indicating that the introduction of too much SiO_2_ amount reduces the thermal stability of alumina aerogels. AS5 also presents a mixed crystal phase of θ-Al_2_O_3_ and α-Al_2_O_3_, but the crystallinity is lower than that of AS4.

### 2.6. SSA and Pore Size Distribution at Different Temperatures

Figure 7a–d show the N_2_ adsorption–desorption curves and pore size distribution of AS0 and AS1 samples after heat treatment from 20 to 1200 °C. It is found that the N_2_ adsorption–desorption curves of the two samples belong to Type-Ⅳ isotherms and the Type-H3 hysteresis loop, which indicates that the two samples mainly possess mesoporous structures formed by the accumulation of nanoparticles [36]. As shown in Figure 7b,d, AS0 and AS1 samples have similar pore size distribution and changing rules. As shown in Figure 8a, in the heating process from 20 to 1200 °C, the specific surface area (SSA) of both AS0 and AS1 gradually increases at first and then decreases slowly, which is caused by the removal of organic groups and the formation of some new mesoporous structure. This point can be identified by the change of sample color and the disappearance of organic groups during the heat treatment. By comparing AS0 and AS1 samples, it is found that AS0 has higher SSA at room temperature, which is mainly due to its lower density and higher porosity. However, in the process of heat treatment, although the SSA of the two types of samples decreases gradually after 600 °C, the reduction rate of AS1 is slow, and the SSA of AS1 exceeds that of AS0 after 1000 °C, which fully shows the promotion effect of silica composite on alumina aerogels to maintain nanoporous structure at high temperature.

Figure 7e,f reflect the N_2_ adsorption–desorption curves and pore size distribution of all the AS samples after heat treatment at 1200 °C. It is found that the N_2_ adsorption–desorption curves of both samples still maintain Type-Ⅳ isotherms and the Type-H3 hysteresis loop. The average pore size of AS0, AS1, AS2 and AS3 is distributed around 30 nm, while the average pore size of AS4 and AS5 decreases rapidly to 20 nm, indicating the sinter of their nanoporous structure. As shown in Figure 8b, the SSA of all AS samples increases at first and then decreases rapidly with the increment of SiO_2_ content, indicating that the appropriate composite amount of SiO_2_ can effectively improve the thermal stability of alumina aerogels. The SSAs of AS2 and AS3 with low SiO_2_ content are up to 283 and 282 m^2^/g, respectively. However, when the composite amount of SiO_2_ exceeds 30%, the SSA of the SiO_2_-Al_2_O_3_ aerogels decreases rapidly, because silica gradually becomes the main skeleton of composite aerogels with low thermal stability, which will inevitably cause the collapse of the skeleton structure at high temperature.

Table 2 shows the average pore size and pore volume of all AS samples. The pore size and pore volume of AS0 and AS1 increase first and then decrease with the increase in temperature, which is consistent with the trend of pore size distribution and specific surface area. In the process of heating the samples from 20 to 600 °C, as the by-products of the reaction and the organic matter attached to the skeleton surface are decomposed at high temperature, leaving more pores, the pore size of AS sample increases, and the specific surface area also increases. In this process, the samples gradually change from grayish brown and brownish yellow to white. Then with the increase in temperature, the SiO_2_ in the samples are sintered, the alumina crystal phase is transformed, the pore size is gradually reduced, the pore volume shrinks and the specific surface area is also gradually reduced. Observing the performance of all AS samples at 1200 °C, it can be seen that AS4 and AS5 are significantly sintered at high temperature due to high SiO_2_ content, and the pore size is significantly reduced. This result is consistent with the data shown above and shows that in the Al_2_O_3_-SiO_2_ composite aerogels, if the Si:Al exceeds 3:7, the high-temperature performance of the aerogels will be seriously reduced.

## 3. Materials and methods 

### 3.1. Materials

Tetraethyl orthosilicate (TEOS), aluminum chloride hexahydrate (AlCl_3_·6H_2_O), ethanol and propylene oxide (PO) were purchased from Sinopharm Chemical Reagent Corporation (Shanghai, China). Deionized water was applied in all experiments. All reagents were analytical grade and were used as received without further purification.

### 3.2. Characterization

The linear shrinkage was calculated depending on the change in the cylinder diameters of the aerogels with an increment in temperature. The samples compared were the as-prepared aerogels derived from the supercritical drying process without any treatment. Bulk density was determined by the mass and volume of regular cylinders. The surface morphology was observed by scanning electron microscopy (ZEISS Sigma 300, Oberkochen, Germany). The chemical groups that remained in the samples were investigated by a Fourier transform infrared spectrometer (FTIR, TENSOR27, Bruker, Karlsruhe, Germany), and all samples were dispersed in dry KBr and pressed into a semitransparent slice for FTIR characterization. The crystal phase of the aerogels was analyzed by powder X-ray diffraction (XRD) with a Rigata/max-C diffractometer using Cu-K_α_ radiation at a scanning speed of 2°/min within the 2 range of 10–90° (Rigaku Ultima IV, Tokyo, Japan). The thermal conductivities were measured by a hotdisk thermal constant analyzer (TPS2500, Sweden). The specific surface area (SSA) and pore size distributions of the samples were obtained by a N_2_ adsorption analyzer (ASAP2460, Shanghai, China). The surface areas and pore sizes were determined by the Brunauer–Emmett–Teller (BET) and Barrett–Joyner–Halenda (BJH) methods.

## 4. Conclusions

In this work, we reduced the molar ratio of alumina to silica gradually from 10:0 to 5:5 to study the influence of the introduction of SiO_2_ on the structure and properties of alumina aerogels. The Al_2_O_3_-SiO_2_ composite aerogels were monolithic, dark brown and opaque without obvious cracks, indicating that they contained some organic groups, which would be gradually removed in the heat treatment process, and the color of the samples eventually turns white. The AS samples with low SiO_2_ content (10~20%) showed excellent thermal performance, which can be identified by their low linear shrinkage of 5%, stable θ-Al_2_O_3_ phase and high SSA of 283 m^2^/g at 1200 °C, because SiO_2_ can hinder the phase transition and collapse of structure effectively. However, when the SiO_2_ content exceeded 30%, the SiO_2_-Al_2_O_3_ aerogels appeared in the α-Al_2_O_3_ phase, leading to a sintering phenomenon and a big loss of SSA to 108 m^2^/g at 1200 °C. Although the skeleton structure of the composite samples was mainly Al_2_O_3_, there were some silica particles and -Al-O-Si- chemical bonds. The -Al-O-Si- bonds entered into the crystal structure of alumina aerogels and hindered their crystal phase transition. These Al_2_O_3_-SiO_2_ composite aerogels derived from a simple, low-priced and safe method with high thermal performance have a wide application in the high-temperature field.

## Figures and Tables

**Figure 1 molecules-28-02743-f001:**
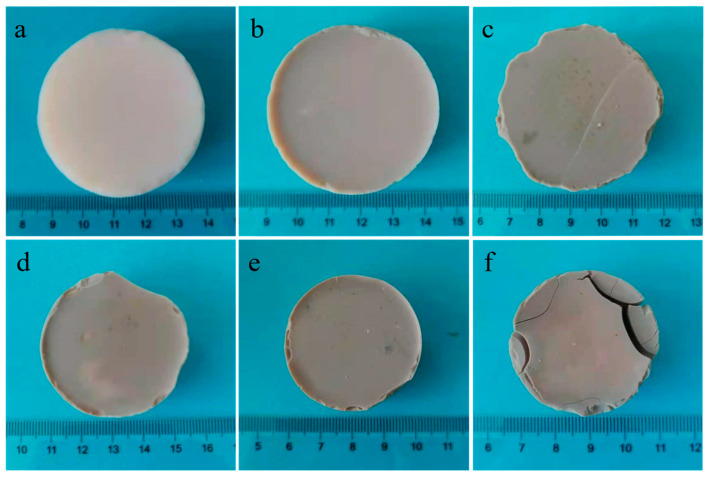
Digital photos of Al_2_O_3_-SiO_2_ composite aerogels (**a**) AS0, (**b**) AS1, (**c**) AS2, (**d**) AS3, (**e**) AS4 and (**f**) AS5.

**Figure 2 molecules-28-02743-f002:**
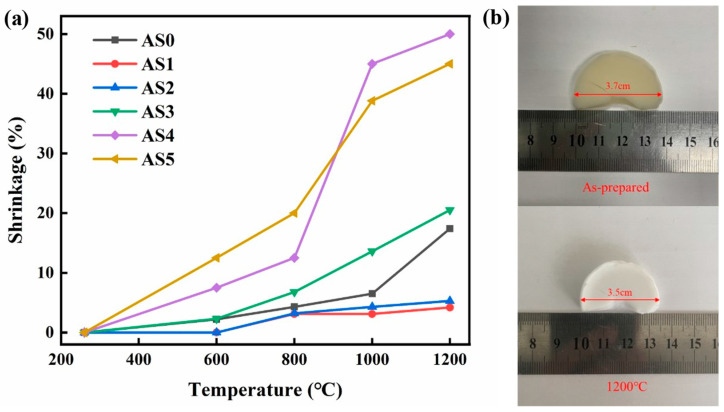
(**a**) The linear shrinkage of AS samples after heat treatment; (**b**) digital photos of AS1 before and after 1200 °C heat treatment.

**Figure 3 molecules-28-02743-f003:**
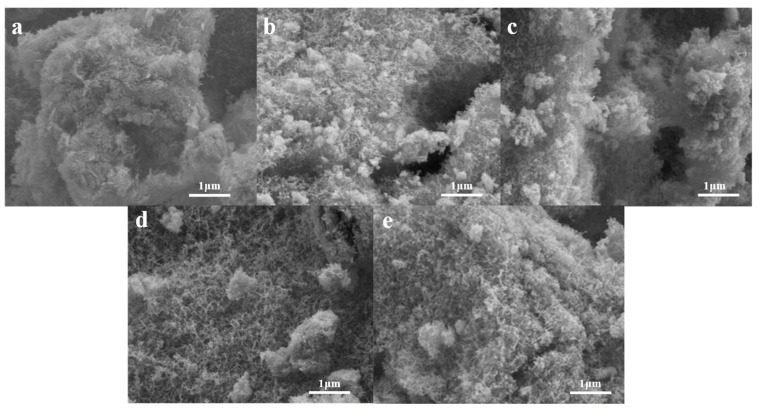
Scanning electron microscopy (SEM) images of AS0 after heat treatment from 20 to 1200 °C: (**a**) AS0-20, (**b**) AS0-600, (**c**) AS0-800, (**d**) AS0-1000, (**e**) AS0-1200.

**Figure 4 molecules-28-02743-f004:**
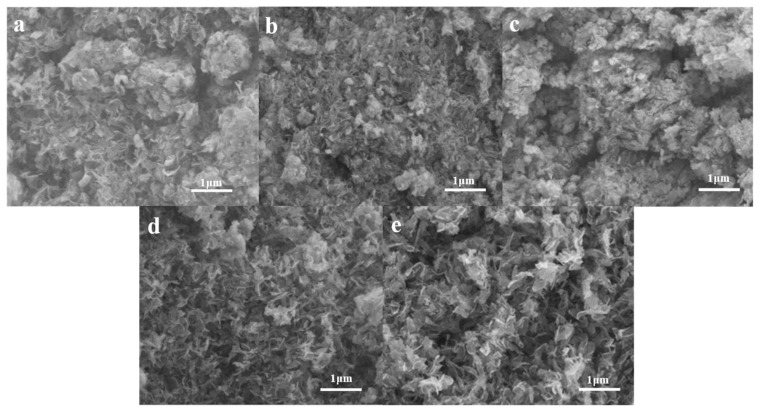
Scanning electron microscopy (SEM) images of AS1 after heat treatment from 20 to 1200 °C: (**a**) AS1-20, (**b**) AS1-600, (**c**) AS1-800, (**d**) AS1-1000, (**e**) AS1-1200.

**Figure 5 molecules-28-02743-f005:**
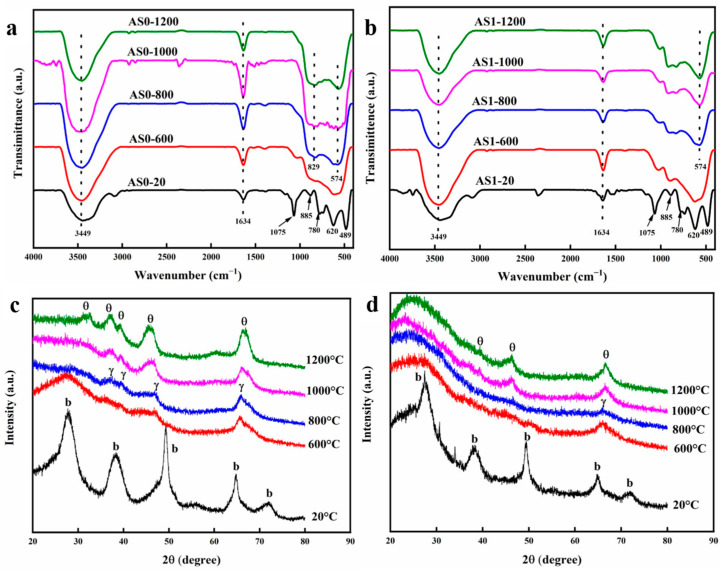
Fourier transform infrared spectrometer (FTIR) of the (**a**) AS0 and (**b**) AS1 after heat treatment from 20 to 1200 °C; X-ray diffraction (XRD) pattern of (**c**) AS0 and (**d**) AS1 after heat treatment from 20 to 1200 °C.

**Figure 6 molecules-28-02743-f006:**
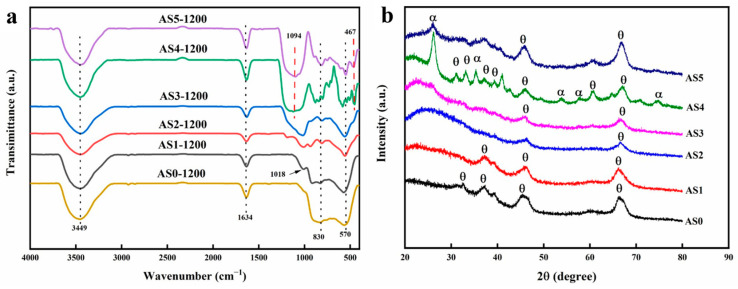
Fourier transform infrared spectrometer (FTIR) (**a**) and X-ray diffraction (XRD) pattern (**b**) of all AS samples after heat treatment at 1200 °C.

**Figure 7 molecules-28-02743-f007:**
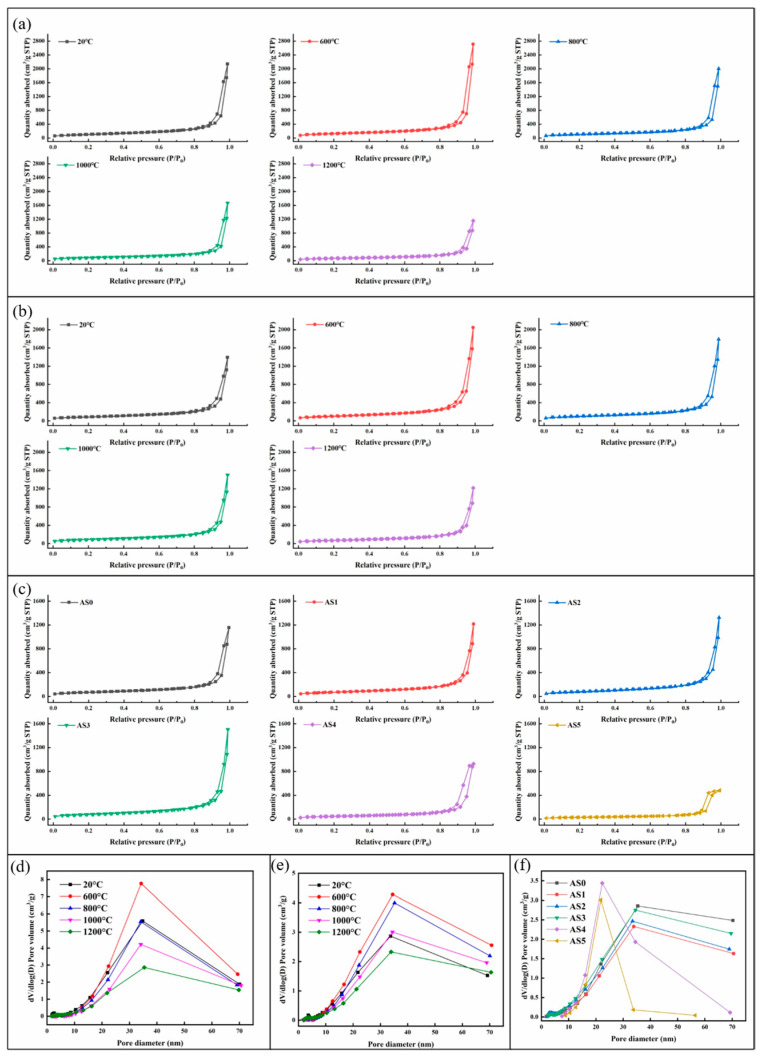
N_2_ adsorption–desorption isotherms of (**a**) AS0, (**b**) AS1 and (**c**) all AS samples after heat treatment at 1200 °C; pore size distribution of (**d**) AS0, (**e**) AS1 and (**f**) all AS samples after heat treatment at 1200 °C.

**Figure 8 molecules-28-02743-f008:**
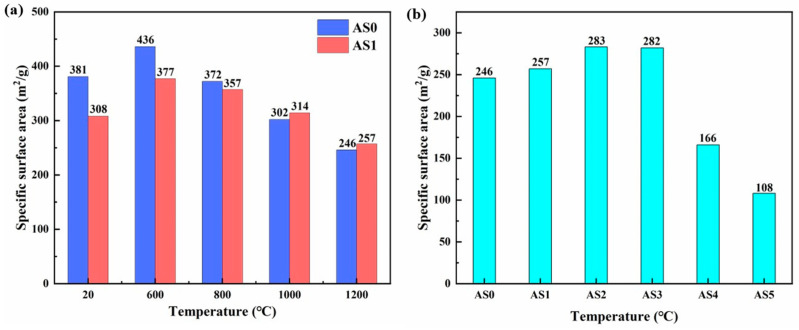
Specific surface area (SSA) of (**a**) AS0 and AS1 samples after heat treatment from 20 to 1200 °C and (**b**) all AS samples after heat treatment at 1200 °C.

**Table 1 molecules-28-02743-t001:** Density, liner shrinkage and thermal conductivity (20 °C) of the AS samples.

Sample	Density (mg/cm^3^)	Liner Shrinkage (%)	Thermal Conductivity (W/m∙K)
AS0	32	12	0.033
AS1	42	8	0.035
AS2	40	10	0.036
AS3	62	15	0.035
AS4	83	23	0.037
AS5	96	23	0.036

**Table 2 molecules-28-02743-t002:** Average pore diameter and pore volume of AS samples.

Sample	Pore Volume (cm^3^/g)	Pore Diameter (nm)	Sample	Pore Volume (cm^3^/g)	Pore Diameter (nm)	Sample	Pore Volume (cm^3^/g)	Pore Diameter (nm)
AS0-20	3.31	31.0	AS1-20	2.15	28.3	AS0-1200	2.30	33.0
AS0-600	4.20	34.2	AS1-600	3.17	33.4	AS1-1200	1.89	29.9
AS0-800	3.09	33.4	AS1-800	2.75	33.6	AS2-1200	2.05	28.8
AS0-1000	2.57	35.6	AS1-1000	2.31	31.8	AS3-1200	2.33	31.5
AS0-1200	2.30	33.0	AS1-1200	1.89	29.9	AS4-1200	1.44	24.6
						AS5-1200	0.75	21.0

## Data Availability

All data are included in the article.

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
