# Peer review of "A Facile Method to Fabricate Al2O3-SiO2 Aerogels with Low Shrinkage up to 1200 °C"

_molecules, 2023, doi:10.3390/molecules28062743_

Round 1
Reviewer 1 Report
This manuscript is about fabrication of Al2O3-SiO2 and the investigation of thermal treatment. The fabrication of the aerogels might be intersting, however, this manuscript is suitable for this manuscript. I recommend the authors will submit another journal. And I have some questions about the manuscript as follows.
1. Experimental should be separated from results. The more details should be shown.
2. The gels contains methyl groups from Si source. Thus, the shrinkage must increase with increase of Si contents. The authors should consider the organic groups by estimation of TG and so on.
3. After the thermal treatment of the aerogels, I think the bulk shapes might not be kept the cyrinder shape. The authors should show the sample photo after thermal treatment. The shrinkage vales are not clear. Please add how to calculate the value.
4. In Line127-159, the description is not clear and the evidence is not enough.
5. Figs. 4 and 6 are not clear. I could not understand what the authors find from the data.
6. The auhors forcus of the thermal properties of their samples. However, the data about the thermal properties could not find. If the authors focus on the thermal properties of the aerogels, I recommed they will estimate the termal properties such as thermal conductivity, thermal expansion and heat resistance of the aerogels.
Author Response
This manuscript is about fabrication of Al2O3-SiO2 and the investigation of thermal treatment. The fabrication of the aerogels might be interesting, however, this manuscript is suitable for this manuscript. I recommend the authors will submit another journal. And I have some questions about the manuscript as follows.
Response: Thanks for the reviewer’s positive comments and efficient work. According to the reviewer’s comments and suggestions, we have revised and improved our manuscript.
- Experimental should be separated from results. The more details should be shown.
Response: Thanks for the comment. We have taken the experimental separately as part 2. We have modified the synthesis process and added some details.
- The gels contains methyl groups from Si source. Thus, the shrinkage must increase with increase of Si contents. The authors should consider the organic groups by estimation of TG and so on.
Response: Thanks for the comment. With the increase of Si content, the amount of methyl introduced by silicon source will also increase, which will indeed lead to larger shrinkage. However, the main purpose of adding Si is to improve the High temperature thermal stability of alumina aerogel. From the results, the incorporation of Si can really achieve the goal. Therefore, we believe that the analysis of the content of methyl and other organic groups will not greatly change the final result.
- After the thermal treatment of the aerogels, I think the bulk shapes might not be kept the cylinder shape. The authors should show the sample photo after thermal treatment. The shrinkage vales are not clear. Please add how to calculate the value.
Response: Thanks for the comment. We have added digital photos of AS1 before and after 1200 ℃ heat treatment. The method for calculating linear shrinkage is described in the Characterization.
- In Line127-159, the description is not clear and the evidence is not enough.
Response: Thanks for the comment. We deleted this paragraph and redescribed the role of Si in alumina aerogels.
- Figs. 4 and 6 are not clear. I could not understand what the authors find from the data.
Response: Thanks for the comment. We rearranged the pictures and uploaded clearer pictures. Figure 5, combined with FTIR and XRD, shows that the crystal phase of AS sample remains stable after high temperature treatment, indicating that its thermal stability is good. Figure 6 shows that AS samples are mesoporous nanomaterials.
- The authors focus of the thermal properties of their samples. However, the data about the thermal properties could not find. If the authors focus on the thermal properties of the aerogels, I recommend they will estimate the thermal properties such as thermal conductivity, thermal expansion and heat resistance of the aerogels.
Response: Thanks for the comment. We have measured the thermal conductivity of all AS samples. It can be seen that the thermal conductivity of AS samples is relatively low and has excellent thermal insulation ability.
Reviewer 2 Report
The manuscript describes the preparation of Al2O3-SiO2 composite aerogels by supercritical drying with ethanol from a mixture containing low-cost aluminum chloride hexahydrate (AlCl3-6H2O) and tetraethyl orthosilicate (TEOS) and propylene oxide (PO).
In its present form, the work cannot be recommended for publication. The manuscript needs some improvements, clarifications and a comprehensive revision.
Some recommendations are included:
ABSTRACT: the best ratio of Al to Si should be shown
INTRODUCTION:
Line 22: high specific surface area
Lines 34, 36, 38, 47: aluminium alkoxides and aluminium inorganic salts instead of alumina alkoxides and alumina inorganic salts: the reactants or alumina precursors are aluminium salts and alkoxides, while the product is alumina.
Line 52: introducing SiO2 instead of compositing. The product is the composite, but regarding the synthesis description Si was added as TEOS in the mixture.
Line 59: structure after microstructure should be removed
SYNTHESIS: the authors wrote: »the solvent of wet gel was displaced with fresh ethanol for 7 times (12h for each time)«. Isn`t this part of the preparation procedure expensive and time consuming?
CHARACTERIZATION of nitrogen physisorption should be described in more details: how was determine the specific surface are, which relative pressure was used to determine specific surface area, which theory was used for pore size determination. Adsorption curve should be used.
RESULTS:
Lines 98, 99, 100 should be removed, they are not part of this manuscript.
Figures are not clear, the sharpness should be improved, also the size of the numbers in Figures 1, 4 and 6. SEM pictures and XRD patterns should be larger.
Table 1: at which temperature the linear shrinkage was determined? This information should be added in the table.
Line 107: what is the meaning of »initial« linear shrinkage?
Lines 127-138
The explanation here should be revised and improved.
Namely, most of the preparations of aluminas (with different crystal structures) are performed by thermal decomposition of alumina hydroxides and oxyhydroxides previously precipitated from solutions containing Al3 + ions or organometallic compounds. Thermal decompositions as well as dihydroxylation (hydroxyl groups are removed under heating) are endothermic processes. Different aluminas can be obtained from 300 to 1200 oC.
The authors should use more recent references regarding explanations of alumina formation.
Lines 145-148
Figure 3 does not show 3D network porous structure, while it shows the surface morphology of the product, which is composed of lamellar and strip nanoparticles. Porosity is present only between particles, which can be seen from the nitrogen isotherms in Figure 5. From SEM it is not possible to see the structure colapse, this can be determined by high temperature XRD. But XRD patterns show only phase transitions at different temperatures. Porous structure can be determined with TEM. It would be good to show TEM pictures of these samples.
Lines 154, 155
The sentence »It can be considered that the introduction of SiO2 makes the micro framework of alumina aerogels coarser, and the strength of its three-dimensional nanostructure is improved.« is not clear and it should be improved. The authors use different names from micro framework, to nanostructure and microstructure. The same designation should be used in the whole text. Nanostructure can be either microstructure or mesostructure, which depends on the average pore size.
Figure 4:
Is there a difference between boehmite and pseudoboehmite?
XRD patterns should contain also all references, for boehmite, and all alumia, which are present. It is difficult to observe diffences in XRD patterns of gamma alumina and theta alumina, because diffraction maxima are very broad.
Discussion on Figure 4C shows that when low content of Si is present (AS1, AS2, AS3) than Si substitutes Al, while higher Si content in AS4 and AS5 leads to formation of SiO2. What is the microstructure of SiO2?
Line 179
Which Al2O3 is present in AS0 at 600C?
Line 239
How much SiO2 is too much?
Figure 6 shows nitrogen adsorption desorption isotherms , not desorption isotherms as it is written.
Line 272 »indicating the collapse of their nano porous structure«
The pore size decrease from 30 to 20 nm is no called the collapse, it can be probably due to sintering of the material.
CONCLUSIONS
Line 288: Authors wrote »because SiO2 can enter the crystal structure of alumina aerogels«
Acording to the results Si (low content) replace some Al atoms in the theta-alumina crystal structure and forms Al-O-Si bonds, while at larger Si contents, SiO2 was formed as Si-O-Si, which cannot enter the crystal structure of the alumina. The authors should improve this sentence.
Line 291 »serious structure collapse« This should be explained.
Line 293: The authors wrote »The -Al-O-Si- bonds not only had strong chemical bond energy«. The authors did not show any results, which confirm chemical bond energy.
Author Response
The manuscript describes the preparation of Al2O3-SiO2 composite aerogels by supercritical drying with ethanol from a mixture containing low-cost aluminum chloride hexahydrate (AlCl3-6H2O) and tetraethyl orthosilicate (TEOS) and propylene oxide (PO).
In its present form, the work cannot be recommended for publication. The manuscript needs some improvements, clarifications and a comprehensive revision.
Response: Thanks for the reviewer’s positive comments and efficient work. According to the comments and suggestions of the reviewers, we revised and improved the manuscript. Revised portions are marked in yellow in the manuscript of highlighted revision..
- ABSTRACT:
The best ratio of Al to Si should be shown.
Response: Thanks for the comment. From the results of linear shrinkage and specific surface area at 1200 ℃, the ratio of Al to Si is 9:1, which is the best ratio. We have added this result to the abstract.
- INTRODUCTION:
Line 22: high specific surface area
Lines 34, 36, 38, 47: aluminium alkoxides and aluminium inorganic salts instead of alumina alkoxides and alumina inorganic salts: the reactants or alumina precursors are aluminium salts and alkoxides, while the product is alumina.
Line 52: introducing SiO2 instead of compositing. The product is the composite, but regarding the synthesis description Si was added as TEOS in the mixture.
Line 59: structure after microstructure should be removed
Response: Thanks for the comment. We have revised the manuscript according to the suggestions of the reviewer and marked the revised part with yellow hightlight.
- SYNTHESIS:
The authors wrote: »the solvent of wet gel was displaced with fresh ethanol for 7 times (12h for each time) «. Isn`t this part of the preparation procedure expensive and time consuming?
Response: Thanks for the comment. Soaking gels in ethanol is the aging process of the gels, which will make the skeleton of gels stronger. When PO is used as proton remover to form gel, by-products will be produced, and PO may not react completely. The purpose of replacing fresh ethanol for many times is to completely remove these products and avoid damaging the skeleton structure of the aerogel during the drying step.
CHARACTERIZATION of nitrogen physisorption should be described in more details: how was determine the specific surface area, which relative pressure was used to determine specific surface area, which theory was used for pore size determination. Adsorption curve should be used.
Response: Thanks for the comment. The specific surface areas were determined by Brunauer-Emmett-Teller (BET) method, and the relative pressure was from 0.05 to 0.30 (P/P0). The pore sizes were calculated by Barrett-Joyner-Halenda (BJH) theory. We have added this part to the Characterization.
- RESULTS:
Lines 98, 99, 100 should be removed, they are not part of this manuscript.
Response: Thanks for the comment. We have removed this part.
Figures are not clear, the sharpness should be improved, also the size of the numbers in Figures 1, 4 and 6. SEM pictures and XRD patterns should be larger.
Response: Thanks for the comment. We have rearranged the figures and submitted clearer figures files as attachments.
Table 1: at which temperature the linear shrinkage was determined? This information should be added in the table.
Response: Thanks for the comment. The diameter of the container which we made the sample is 5.6cm. Here, we calculate the linear shrinkage of the dried sample compared with the wet gel.
Line 107: what is the meaning of »initial« linear shrinkage?
Response: Thanks for the comment. The “initial linear shrinkage” refers to the linear shrinkage after drying. We are sorry we use the wrong words here. We have revised the words
Lines 127-138 The explanation here should be revised and improved.
Namely, most of the preparations of aluminas (with different crystal structures) are performed by thermal decomposition of alumina hydroxides and oxyhydroxides previously precipitated from solutions containing Al3+ ions or organometallic compounds. Thermal decompositions as well as dihydroxylation (hydroxyl groups are removed under heating) are endothermic processes. Different aluminas can be obtained from 300 to 1200 oC.
The authors should use more recent references regarding explanations of alumina formation.
Response: Thanks for the comment. Alumina aerogel is formed by hydrolysis and polycondensation of aluminum ions. This aerogel is a nano porous material, and there will be a large number of hydroxyl residues on the surface of its internal framework. The model used here is to better explain the sintering phenomenon of alumina aerogel during heating. We are sorry to write endothermic as exothermic and we have corrected it.
Lines 145-148:
Figure 3 does not show 3D network porous structure, while it shows the surface morphology of the product, which is composed of lamellar and strip nanoparticles. Porosity is present only between particles, which can be seen from the nitrogen isotherms in Figure 5. From SEM it is not possible to see the structure colapse, this can be determined by high temperature XRD. But XRD patterns show only phase transitions at different temperatures. Porous structure can be determined with TEM. It would be good to show TEM pictures of these samples.
Response: Thanks for the comment. We are sorry that we cannot easily retest a copy of TEM data due to equipment maintenance. However, Figure 3 and Figure 4 show the morphology and structure of the sample after heat treatment at different temperatures. Compared with the change of pore size distribution and specific surface area, it can be seen that after heat treatment, the pore size decreases and the specific surface area decreases, which shows that the samples are porous structure.
Lines 154, 155:
The sentence »It can be considered that the introduction of SiO2 makes the micro framework of alumina aerogels coarser, and the strength of its three-dimensional nanostructure is improved.« is not clear and it should be improved. The authors use different names from micro framework, to nanostructure and microstructure. The same designation should be used in the whole text. Nanostructure can be either microstructure or mesostructure, which depends on the average pore size.
Response: Thanks for the comment. Aerogels are mainly mesoporous materials, which can be seen from the average pore size distribution. We have corrected the words in the full text.
Figure 4:
Is there a difference between boehmite and pseudoboehmite?
Response: The chemical formula of boehmite and pseudo boehmite are identical, but the crystallization of boehmite is regular, and the crystallization of pseudo boehmite is amorphous.
XRD patterns should contain also all references, for boehmite, and all alumia, which are present. It is difficult to observe differences in XRD patterns of gamma alumina and theta alumina, because diffraction maxima are very broad.
Response: Thanks for the comment. We have marked all obvious peaks in the XRD patterns. The purpose of the XRD patterns used here is to find out whether there is É‘-Al2O3 in the aerogel after high-temperature treatment, because the É‘-Al2O3 is the most closely arranged crystalline phase of alumina. When marking crystal phase, we only marked the position of the peak tip, because the diffraction peaks in the graph are very wide. However, since these positions are different from the common É‘-Al2O3 peak positions, it is sufficient to explain the thermal stability of aerogels.
Discussion on Figure 4C shows that when low content of Si is present (AS1, AS2, AS3) than Si substitutes Al, while higher Si content in AS4 and AS5 leads to formation of SiO2. What is the microstructure of SiO2?
Response: Thanks for the comment. The silicon source used for AS sample is TEOS, which can form silica hydrogel through hydrolytic condensation. The microstructure of silica aerogel is similar to that of alumina aerogel, but since silica will decompose at above 650 ℃, this will lead to sintering.
Line 179
Which Al2O3 is present in AS0 at 600C?
Response: Alumina is mainly boehmite phase at 600 ℃.
Line 239
How much SiO2 is too much?
Response: From the results of this work, the thermal stability of AS4 and AS5 samples with lower silicon content is significantly decreased compared with that of AS5 samples. It can be considered that the silicon content is too high when the molar ratio of silicon to aluminum exceeds 3:7.
Figure 6 shows nitrogen adsorption desorption isotherms, not desorption isotherms as it is written.
Response: Thanks for the comment. We have corrected it.
Line 272 »indicating the collapse of their nano porous structure«
The pore size decrease from 30 to 20 nm is no called the collapse, it can be probably due to sintering of the material.
Response: Thanks for the comment. The reduction of the average pore size from 30 nm to 20 nm is probably caused by sintering. We have revised the statement.
CONCLUSIONS
Line 288: Authors wrote »because SiO2 can enter the crystal structure of alumina aerogels«
Acording to the results Si (low content) replace some Al atoms in the theta-alumina crystal structure and forms Al-O-Si bonds, while at larger Si contents, SiO2 was formed as Si-O-Si, which cannot enter the crystal structure of the alumina. The authors should improve this sentence.
Response: Thanks for the comment. We have revised the statement.
Line 291 »serious structure collapse« This should be explained.
Response: Thanks for the comment. I'm sorry that we made a writing error here. It should have been "structural collapse". According to your suggestion, we think it may not be structural collapse but sintering phenomenon. We have revised the statement.
Line 293: The authors wrote »The -Al-O-Si- bonds not only had strong chemical bond energy«. The authors did not show any results, which confirm chemical bond energy.
Response: Thanks for the comment. We are sorry that the relevant content is not mentioned in the article, so we have deleted the sentence.

Round 2
Reviewer 1 Report
This manuscript is about fabrication of Al2O3-SiO2 aerogels. However, the aims of this manuscript is not clear. The authors should add the aim of this work in introduction section. Beause the aim is not clear, I could not understand well what the authors found in this work. I recommend the authors should improve the manuscript before acceptance. The details are shown as follows.
1. In introduction, the authors wrote that the influence of SiO2 content on the heat resistance were studied. However, the author did not explain the influence on the heat resistance. The authors should show the results about heat resistance.
2. Why did all samples look like gray? The color depended with increase of SiO2 content. I could not understand why colora changed. How did the shape and color of the samples changed after sintering?
3. In line 111, the adding of a small amount of SiO2 improves the thermal stability. The shrinkage does not represent the thermal stability. I could not understand what the sentence means. It si not clear why AS4 and AS5 shrunk in the range from 800 to 1000 deg. Did the density of the samples after sintering estimate after sintering?
4. Please consider the phase diagram of Al2O3-SiO3 and estimate to the XRD patterns after sintering.
5. From the N2 adsorption/desorption curves, the information of pores such as pore volume and so on should be inversitageted. The curve was assigned to Type H3 but alomost of the samples looks like Type II. In that case, the pore size are more than 50 nm and the pore distribution might be explained. The N2 adsorption/desorption curves should be estimated more precisely. And the data should be shown in the manuscript such as Table.
Reviewer 2 Report
The authors have addressed all the recommendations.
One small improvement should be added. The authors should specify the temperature at which the thermal conductivity measurements were made. This is important information because thermal conductivity depends on temperature.
After this minor revision the manuscript can be published.
Author Response
Reviewers’ comments:
Comments and Suggestions for Authors.
The authors have addressed all the recommendations.
One small improvement should be added. The authors should specify the temperature at which the thermal conductivity measurements were made. This is important information because thermal conductivity depends on temperature.
After this minor revision the manuscript can be published.
Response: Thank you very much for your affirmation of the manuscript. The thermal conductivities given in the manuscript was measured at 20 ℃, and we have added this information to the data.